# Recent Large-Scale Genotyping and Phenotyping of Plant Genetic Resources of Vegetatively Propagated Crops

**DOI:** 10.3390/plants10020415

**Published:** 2021-02-23

**Authors:** Hilde Nybom, Gunārs Lācis

**Affiliations:** 1Department of Plant Breeding–Balsgård, Swedish University of Agricultural Sciences, Fjälkestadsvägen 459, SE-29194 Kristianstad, Sweden; 2Institute of Horticulture, Graudu 1, LV-3701 Dobele, Latvia; gunars.lacis@llu.lv

**Keywords:** DNA marker, genetic diversity, GWAS, pedigree, plant breeding, SNP, SSR

## Abstract

Several recent national and international projects have focused on large-scale genotyping of plant genetic resources in vegetatively propagated crops like fruit and berries, potatoes and woody ornamentals. The primary goal is usually to identify true-to-type plant material, detect possible synonyms, and investigate genetic diversity and relatedness among accessions. A secondary goal may be to create sustainable databases that can be utilized in research and breeding for several years ahead. Commonly applied DNA markers (like microsatellite DNA and SNPs) and next-generation sequencing each have their pros and cons for these purposes. Methods for large-scale phenotyping have lagged behind, which is unfortunate since many commercially important traits (yield, growth habit, storability, and disease resistance) are difficult to score. Nevertheless, the analysis of gene action and development of robust DNA markers depends on environmentally controlled screening of very large sets of plant material. Although more time-consuming, co-operative projects with broad-scale data collection are likely to produce more reliable results. In this review, we will describe some of the approaches taken in genotyping and/or phenotyping projects concerning a wide variety of vegetatively propagated crops.

## 1. Introduction

Propagation method has profound effects on both plant cultivation and research. Most of the vegetatively propagated crops are perennial, cross-pollinated, and often self-sterile. There is sometimes a progression from seed propagation of landraces to vegetative propagation of more actively selected or intentionally bred cultivars. Seeds derive from crosses between two heterozygous parents and are not genetically identical to the mother plant. Multiplication of selected genotypes with valuable traits is therefore achieved only through vegetative propagation methods. This also affects the research, breeding, and preservation of valuable germplasm, which must be focused on particular genotypes instead of on seedling families as in seed-propagated crops. Genetic research and development of whole genome sequences is further complicated by the fact that these crops are often polyploid (e.g., banana, mango, ornamental rose, potato and strawberry) or derive from whole-genome duplication, so-called ancient polyploids (apple and pear).

Especially desirable genotypes of, e.g., apple and grapes, have probably remained unaltered for hundreds of years and have been disseminated across large areas, with profound effects on the amount and distribution of genetic variation. Clonal propagation of long-lived crops also involves the emergence of somatic mutants or “sports”, some of which represent an improvement of a particular trait. In some crops like grape, the growing of local varieties in restricted locations like mountain valleys has created locally adapted clones, which have often spread further under different names [1]. The true affiliation of these accessions can only be determined through genotyping. Distinct and valuable clones must be preserved to act as a source of propagation material, but the transmission of traits to the offspring is usually identical to that of the original genotype.

### 1.1. Preservation of Plant Genetic Resources

Fruit and nut trees are traditionally conserved in ex situ field collections as budded or grafted cultivars, while other trees, bushes, and large herbs are grown from leaf or stem cuttings, or from suckers harvested at the stem basis. Field-planted collections are required also for tuber crops like potatoes, sweet potatoes and cassava, although tubers may be preserved in cold storage for a limited time. The collections are hosted by a wide range of governmental organizations, universities, private companies, botanical gardens, foundations, public parks and individuals. Many publicly funded collections are focused on the preservation of heritage accessions and include mostly old and/or local cultivars from a defined geographic area. Other collections are designated mainly for genetic studies and plant breeding, and may include both old and new cultivars from various areas, advanced breeding lines and wild relatives. Field collections are sometimes backed up by in vitro cultures and cryopreserved tissue (buds and shoot tips), whereas older and/or local cultivars are sometimes preserved in situ (fields and orchards).

The European Cooperative Program for Genetic Resources (ECPGR) has created a network of plant conservationists and scientists, with working groups designated to specific crops, e.g., berries, *Malus*/*Pyrus*, potato, *Prunus* and *Vitis*. The Global Crop Diversity Trust has employed specialists to develop global ex situ conservation strategies for some key crops. Thus, Bramel and Volk [2] outlined a strategy for apple, which could be useful also for other vegetatively propagated crops: (1) coordinate a global genotyping effort to assess and map diversity within and among accessions in the existing global collections, verify accession identity, confirm redundancies, identify mislabeled accessions, identify unique accessions, and determine the existence of key gaps in the global collections and (2) ensure access to an information platform with an international registry and database for ex situ and in situ collections, including tree- and accession-level phenotype, passport, and genotype information.

Global germplasm exchange is facilitated by the International Treaty on Plant Genetic Resources for Food and Agriculture (ITPGRFA), which aims to strengthen the conservation, exchange, and sustainable use of plant genetic resources for food and agriculture. Several public databases for accession-level information sharing are available like the European Search Catalogue for Plant Genetic Resources (EURISCO) and the GRIN-Global database for accessions conserved by the USDA-ARS National Plant Germplasm System. Genesys, initiated by Bioversity International, the Global Crop Diversity Trust, and the secretariat of ITPGRFA, was launched as a global platform in 2011, with cooperation by EURISCO, GRIN, and the System-Wide Information Network for Genetic Resources (SINGER) from CGIAR. Not all crops are, however yet covered by these databases and the information is fast becoming obsolete since accessions continuously enter or leave different plant collections.

### 1.2. Accession-Level Information

Each preserved accession in a plant collection should have passport data (accession name, genus name, country of origin, acquisition date, code of the institute where the accession is maintained, and a unique accession number). The collection becomes infinitely more valuable if the accessions are accompanied by phenotypic data (descriptors and traits of economical relevance) and data from molecular studies on true-to-type status and markers/genes for traits of interest. Many plant collections were established long before DNA fingerprinting became available. Decisions about what accessions to preserve have therefore relied mainly on morphological traits, information on ecogeographical origin and/or historical importance. The number of redundant (synonymous genotypes) and mislabeled accessions frequently reaches 30% or even higher when these collections are investigated with molecular markers [3].

In order to increase the importance and utilization of the preserved accessions, co-ordinated national and regional efforts are needed to compare collections at the phenotypic and genetic levels, and to confirm cultivar identity and identify valuable genes. The scientific community has produced considerable amounts of data for accessions in various plant collections. Several rosaceous fruit and berry crops were thus investigated in recent multinational research projects; the RosBREED projects in 2009–2019 involved many research institutes in the USA and some overseas [4]. Another major project was the European Union-funded FruitBreedomics project (2011–2015) involving 28 research institutes and private companies working with apple and/or peach [5].

## 2. Genotyping

Of all DNA fingerprint methods emerging in the early 1990s, only simple sequence repeats (SSR, also known as microsatellites) remain as a major option for screening collections of plant genetic resources, although other single-locus methods like SCoT (start codon targeted) markers, and multilocus methods like ISSR (inter-simple-sequence repeats), IRAP (inter-retrotransposon amplified polymorphism), REMAP (retrotransposon-microsatellite-amplified polymorphism), and SRAP (sequence-related amplification polymorphisms) are occasionally applied. Large-scale analyses are sometimes also carried out with the diversity arrays technology (DArT), which is based on the hybridization of fluorescent DNA probes to target DNAs spotted onto a microarray.

However, with the arrival of next-generation sequencing (NGS) technologies, a rapidly increasing number of studies have instead utilized single nucleotide polymorphisms (SNPs) and various other sequencing-based technologies like DArTseq.

### 2.1. Simple Sequence Repeats (SSR)

SSR have long been the preferred marker for genotyping vegetatively propagated plants, mainly because of their multiallelic and highly polymorphic nature, high discriminatory potential, and good reproducibility [3]. Once the primers for suitable SSR loci are developed, laboratory work and data analysis are straight-forward. A definite bonus is that additional samples can be cost-effectively screened and added into an already existing database for renewed analyses of diversity and relatedness. Combining datasets from multiple laboratories can, however, become problematic due to discrepancies in allele size evaluations, but adjustments are facilitated by comparison of shared reference accessions, sometimes complemented with the regenotyping of a subset of accessions [6].

SSR-based analyses usually report very high levels of interaccession polymorphism, and duplicate samples are easily detected. By contrast, somatic mutants, so called sports or clones, are seldom distinguished from their original genotypes although clone-specific mutations are occasionally identified [7]. More than two alleles in the same locus/sample usually indicate polyploidy; 10–30% triploids are commonly reported from germplasm screenings of apple and pear [8,9,10,11,12,13]. SSR analyses can also indicate which one of two diploid parental genotypes has produced an unreduced gamete resulting in a triploid offspring. Supernumerary bands can, however, also derive from duplication events or somatic mutations generating a chimerical state [10].

Comparison of results across studies is often hampered by the use of different sets of SSR loci [14]. To remedy this situation, ECPGR has published several sets of recommended SSR loci, e.g., one set for pear [15], one for cherry [16], and one for plum [17]. For pear, a different set of loci (the US *Pyrus* Genetic Resources set, USPGR) was recently proposed, with longer repeat motifs (3–6 bp) resulting in fewer alleles but also less artifacts such as stutters, split peaks, and binning errors, which are common when using dinucleotide repeat loci as in the ECPGR set [18]. The ECPGR set for apple has been modified over the years as more information about polymorphism and linkage has become available, and now includes 17 SSR loci that span most of the apple genome [6].

A standardized and internationally recognized set of SSR loci is used not only for the coordinated characterization of genetic resources, but also as tools for cultivar identification and protection of Plant Breeders’ Rights. Such SSR marker-based genotype data are included in the International Variety Catalog (VIVC) for grapes [19] and The European Union Common Catalog (EUCC) for potato [20].

### 2.2. Single Nucleotide Polymorphisms (SNPs)

Several approaches are used to generate SNPs in plant material. Genotyping-by-sequencing (GBS) produces SNPs from the whole genome, and is especially valuable for plants where reference genome sequences have not yet been published. One way to reduce the complexity in GBS is RAD-Seq (restriction-site associated DNA sequencing). Using this technique, 1.25 million SNPs were discovered in a panel of 84 mango varieties and used for development of the genomic resource MiSNPDb [21]. SNPs are also commonly mined from expressed sequence tag (EST) or transcriptome gene sequences. Insertion/deletion variants occur frequently and are sometimes also used as markers but their reproducibility is generally lower than for SNPs.

Use of a chip or microarray can reduce time and cost for genotyping large sets of accessions. SNP arrays have been developed for many commercially important crops, such as the 6K cherry [22], 8K apple [23], 9K peach [24], 18K grape [25], and 6K avocado arrays [26] implemented on the Illumina Infinium^®^ or Illumina II^®^ platforms. Similarly, a 20K array was developed for apple [27] but the screening of a diverse germplasm indicated very rapid decay of the linkage disequilibrium (LD) for each apple chromosome. A number of markers had to be increased to obviate this problem, and the Axiom^®^ apple 480K SNP array was subsequently developed [28]. Several other Axiom^®^ arrays have been developed, including a 20K array for potato [29], a 68K array for rose [30], a 700K array for walnut [31], and both a 70K array and a 200K array for pear [32,33]. For the allo-octoploid strawberry, a 90K array has been produced [34] and both a 50K “production” array and an 850K “screening” array, with SNPs chosen according to the different purposes [35].

The SNPs must be developed from a genetically relevant material to avoid a biased representation and thus failure to reflect true differentiation among investigated samples. SNP discovery panels can differ considerably between projects aiming to produce a genomic map [26] and projects aiming to analyze genetic variation and identify marker-trait associations in germplasm collections [27]. Ascertainment bias is probably more common in SNP arrays compared to GBS due to the use of biallelic SNPs and lack of sufficient diversity in the discovery panels. Conversely, GBS often involves a higher amount of missing data, and sequence coverage must be very high to achieve reproducibility between different sets of experiments [18].

Highly automated scoring of SNPs can easily generate errors that affect the deductions made from obtained results. A common workflow (using software ASSIsT) was therefore designed for removal of faulty SNPs in apple, peach, and sweet cherry, according to inheritance principles and pre-existing pedigree information [36]. SNP-based analyses can also help to identify chromosome number levels. Theoretically, triploid individuals should have a 50% higher level of heterozygosity than diploid individuals, assuming Hardy–Weinberg equilibrium conditions. The difference was, however, only 30% in a collection of apples, probably due to the lack of a proper equilibrium across the investigated samples [37].

It has been argued that use of GBS is overall less expensive than SNP arrays since marker discovery and genotyping can be achieved in a single step in highly diverse crops like apple [37], and can be applied also in crops for which a reference genome is still lacking. For smaller projects, SSR is still cheaper since a low number of samples can be genotyped whereas at least 96 samples are needed for cost-effective GBS and SNP array genotyping [18].

## 3. Genetic Diversity

Assessments of diversity can be used to reveal genetic identity (synonyms, homonyms, and mislabeling) and genetic relationships (parentage and kinship, sports, and hybrids) among samples, thereby facilitating both management and use of the preserved accessions. High levels of polymorphism in, e.g., ancient germplasm indicate the presence of genes that could be valuable in the face of climate change and new market demands. Changes in polymorphism over time may reflect the existence of, e.g., genetic bottlenecks during domestication. Evaluation of genetic diversity and population structure also yields valuable information for plant breeding-related research such as genome-wide association studies (GWAS) and genomic selection (GS).

### 3.1. Quantification of Variability and Structuring

Analyses of genetic diversity in one or several plant collections can provide valuable information, not just on the collections but also on how the various crops have been affected by biological and historical factors such as breeding system (self- or crosspollinated), plant longevity, distribution area, domestication history, and orchard/field management. Vegetatively propagated crops are often characterized by high diversity among accessions due to, e.g., obligate cross-fertilization, multiple hybridization events combined with human activities like selection and breeding, and diversifying selection associated with adaptation to different agricultural environments and uses. By contrast, geographic differentiation is often low since vegetative propagation has enabled dispersal of selected genotypes over wide areas.

Many statistical analysis methods are being used but only a few will be mentioned here. SSR data are especially difficult to handle in polyploid crops, but software like SPAGeDI 1.3 [38] and PolySat [39] allows the use of average allele size as a replacement for “missing alleles” in loci where fewer than the possible maximum number of alleles are scored. Genetic structure in the studied collection and differentiation among groups of accessions is usually investigated with the Bayesian model-based cluster procedure [40], which shows the extent to which each individual genotype is related to one or several not predefined genomic groups. Discriminant analysis of principal components (DAPC) is similar to the Bayesian structure analysis but does not require a Hardy–Weinberg equilibrium and is therefore well suited for polyploids. Genetic differentiation among predefined groups is often quantified with analyses of molecular variance (AMOVA) using, e.g., the GenoType/GenoDive package [41], which can handle varying ploidy levels. Relationships among genotypes are often illustrated by hierarchical clustering but the resulting dendrograms are seldom able to properly reflect hybridization events. Principal component analysis or coordinate analysis (PCA or PCoA) and factorial correspondence analysis (FCA) can often provide a more accurate overview.

Sequencing data are also difficult to handle in polyploids, and specialized software may be required like EBG with two models (for autopolyploids and for allopolyploids, respectively), which was recently developed for estimating genotypes and population genetic parameters [42].

#### 3.1.1. The Family Rosaceae

Apple: Cultivated apple belongs to the hybridogenous *Malus* × *domestica* Borkh., derived mainly from the Central Asian *M*. *sieversii* (Ledeb.) M.Roem. but with some input also from the European *M*. *sylvestris* Mill. Numerous national germplasm collections have been screened using SSR loci; e.g., Urrestarazu et al. [6] investigated 493 mostly old and local accessions maintained in three collections in Northeastern Spain, Liang et al. [10] investigated 418 mainly Italian accessions of which only 192 were shown to be unique, Gaši et al. [43] investigated 181 accessions derived from 6 collections in Norway, and Lassois et al. [11] investigated 2163 accessions sampled from several French collections showing 34% redundancy.

In the hitherto most wide-ranging study, a common set of 16 SSR loci were used to genotype more than 2400 (mainly old local/national) accessions from 14 collections in 9 European countries, Western Russia, and Kyrgyzstan [44]. A total of 405 accessions were removed due to redundancy both within and between collections despite being labeled with unique names. The extensive diversity in the remaining set of cultivars was organized in three moderately differentiated groups according to their area of origin in Europe: North + East (141 genotypes), West (775 genotypes), and South (148 genotypes), suggesting an influence of historic factors as well as climate adaptation. A finer delineation into eight subgroups reflected both the geographic origin and usage like cider production.

Some studies have included wild apple relatives for comparison. Thus Gross et al. [45] used 9 SSR loci to investigate 1228 genotypes of cultivated apple together with some samples of the progenitor species *M. sieversii*, *M. sylvestris,* and *M. orientalis* Uglitzk. preserved by the Plant Genetic Resources Unit at USDA, Geneva, USA. A subset of these were subsequently analyzed with 10 additional SSR loci. Diversity estimated as expected heterozygosity (He) and allelic richness was somewhat lower in cultivated apple compared to each of the wild species. Determination of the diversity for 735 cultivars developed at various times (from the 13th century up until now) showed no significant reduction in variability through time during the last eight centuries. In another study based on the USDA apple collection, 1063 diploid, unique accessions of *M*. × *domestica*, *M*. *sieversii*, *M*. *baccata* (L.) Borkh., *M*. *floribunda* Siebold ex Van Houtte, *M*. *orientalis,* and *M*. *sylvestris* were investigated with almost 28,000 GBS-generated SNPs (Migicovsky et al. 2021). A Bayesian genetic structure analysis produced three groups, consisting mainly of *M*. *domestica*, *M*. *sieversii,* and *M*. *floribunda* + *M*. *baccata*, respectively. Dessert apples were shown to have a relatively higher influence of *M*. *sieversii* compared to cider apples, which instead showed more influence of *M*. *sylvestris*.

Two different genotyping methods were compared by Larsen et al. [37]; 15 SSR loci and 15,000 SNPs obtained with GBS were used to study the genetic diversity of 272 unique apple accessions in the Pometum gene bank collection, Copenhagen, Denmark. The analyses yielded overall concordant results but the SNPs provided a more detailed population structure.

Several kinds of array-based methods have been applied, e.g., DArT, which was used to study genetic diversity and pedigree relationships in over 2000 accessions in the National Fruit Collection (NFC) at Brogdale in Great Britain [12]. Instead using SNP, 215 accessions from the Swedish Central collection focused on heritage cultivars, in Alnarp, Sweden, were genotyped with the 20K apple Illumina Infinium^®^ SNP array [46] while more than 1400 accessions in six European collections were analyzed with the Axiom^®^ apple 480K SNP array [47].

Pear: Pear fruits have been produced commercially for centuries, and are divided into “Asian pears” (several species in the genus *Pyrus*) or “European pears” (*P. communis* L.). Liu et al. [48] used 134 SSR loci to genotype 385 pear accessions of five species preserved at the Chinese National Pear Germplasm Repository in Wuhan, China. Differentiation among species was low but European pear accessions separated clearly from the Asian pears. Substructuring of the material provided information on putative taxonomic relationships, geographic origin of taxa and historical gene flow. Similar results were later obtained when 10,186 GBS-derived SNPs were used to investigate 231 pear accessions from 16 countries preserved at the National Germplasm Repository in Korea [49]. European and Asian pears were again divided into two separate groups, and the Asian pears were further subdivided according to taxonomic origin.

In-depth analyses have been carried out with more local germplasm, as the three European pear collections preserved in North-Eastern Italy, where pears have been grown for a very long time [8]. The samples were analyzed with some of the ECPGR-approved SSR loci, and high levels of polymorphism were encountered among the 120 unique accessions. A large proportion of the accessions were identified as cooking and processing cultivars, thus reflecting utilization in past centuries.

SNPs proved to yield the most robust results when a set of 1416 pear accessions preserved at USDA-ARS in Corvallis, USA, were genotyped with the 70K SNP Axiom^®^ array and with GBS with restriction enzyme-reduced representation of sequences [32]. In a second study, analysis with this SNP array was compared to analyses conducted with two different sets of SSR loci (10 USPGR loci and 12 ECPGR loci, respectively) for ability to assess diversity, population structure, pedigree, and identity. All three methods produced similar results, but genotyping with the USPGR SSR set was regarded as the most user-friendly alternative [18]. In a third study of 1890 diploid *Pyrus* spp. samples investigated with the 70K SNP Axiom^®^ array, a major dichotomy was again found between *P. communis* on the one hand and the Asian pear species on the other hand, and with several subgroups within each major group [50].

Stone fruits: Stone fruits belong to the genus *Prunus* and have a long history of cultivation and are widespread in all regions of the world. These species are characterized by a relatively small genome (250–300 Mbp) with eight chromosomes at the basic level and high synteny [51]. Genome sequence information and molecular markers developed for one Prunus species are often easily transferred to other species, facilitating research in the genus. SSR markers are widely used in *Prunus* germplasm research [52], and provide detection of identical genotypes, assessment of variability, and grouping of accessions into species like peach (*P. persica* (L.) Batch), European plum (*P. domestica* L.), apricot (*P. armeniaca* L.), sweet cherry (*P. avium* L.), and almond (*P. dulcis* (Mill.) D.A.Webb).

The peach genome was the first to be sequenced (including repeated updates) [53,54], which has enabled development of a number of SNP marker arrays, annotation of sequences, and identification of genes. Peach evolution, genetic changes during domestication and breeding, and associations with agronomic traits was studied in 480 peach accessions (improved cultivars, landraces, and wild samples) using SNP identification, annotation, and genotyping [55]. In another study, 775 high-quality SNPs were selected from more than 16 million putative SNPs to analyze genetic relationships among 360 peach accessions [56]. The Illumina Infinium^®^ peach 9K SNP array developed by the International Peach SNP Consortium (IPSC) was used for the hitherto most comprehensive peach diversity analysis with 1580 accessions [57]. Domestication and cultivar differentiation were studied using the same array in 95 accessions including peach cultivars, peach relatives (*P. davidiana* (Carr.) Franch., *P. kansuensis* Rehd., *P. mira* Koehne, *P. tangutica* (Batalin) Koehne, and *P. webbii* (Spach) Vierh.) and outgroup species like *P. salicina* Lindl., *P. mume* Siebold and Zucc., and *P. avium*, and demonstrated the transferability of genetic information among different *Prunus* species [58].

Several national and international SSR-based genotyping projects in hexaploid European plums have provided an overview of genetic structuring and the grouping of accessions into taxonomic and/or pomological classification systems [59,60]. Using GBS-derived SNPs in a study on 405 accessions, low diversity of cultivated plum was reported and a limited number of founder cultivars, confirmation of the origin of hexaploid cultivated plums from the wild progenitor species *P. cerasifera* Ehrh. and *P. spinosa* L., and the evolutionary development of *P. cerasifera* [61].

Several SSR-based analyses have investigated diversity in local sweet cherry genetic resources [62,63,64,65], sometimes complemented by markers developed from the self-incompatibility (*S*) locus sequence [66]. Development of the latest 6+9K SNP array for sweet and sour cherries will facilitate the characterization of cherry germplasm [67].

Berry crops: Although strawberry (*Fragaria* × *ananassa* Duchesne) is one of the most widely grown horticultural crops, genetic research has lagged behind due to the high ploidy level (2*n* = 8x = 56). Over time, a large range of molecular markers has been used for diversity studies [68]. A breakthrough was provided by the development of a 90K Axiom^®^ SNP array [34], which was applied to 384 individuals including samples of the diploid relatives *F. vesca* L. and *F. iinumae* Makino to create a road-map for SNP development and screening in high-ploidy species. This SNP array has also been used for cultivar discrimination [69], mapping of agronomically significant traits, marker development, and genome-wide prediction [70]. Related diploid species can be used as a model for studying the heredity of traits, and act as a potential source of useful genes. The natural range of *F. vesca* and its subspecies include the entire northern hemisphere. Nearly 300 genotypes from 274 locations in 31 countries were collected and genotyped using 68 SSR loci [71]. Comparatively low levels of genetic diversity were reported compared to other Rosaceae species, possibly explained by differences in reproductive features. Genetic differences between Eurasian and American samples were identified, although they did not differ significantly in morphology. A set of 24 SSR loci, based on sequences of genes involved in regulation of biosynthesis and plant–pathogen interactions in *Fragaria*, were successfully applied to both *Fragaria* and *Rubus* germplasm (including rare cultivars and wild relatives) thus ensuring transferability of markers across genera [72].

The genus *Rubus* contains several crops with a significant place in fruit production: diploid raspberries, polyploid blackberries, and numerous interspecific hybrids. The genetic base and its use in breeding were addressed by genotyping the German *Rubus* collection with 16 SSR loci [73]. Problems with a restricted gene pool have been pointed out based on data from various molecular markers [74]. One way to increase genetic diversity is to involve wild relatives in the breeding. Population structure and variability has therefore been studied with 23 SSR loci in accessions of, e.g., *R. coreanus* Miq., *R. crataegifolius* Bunge, *R. crataegifolius* var. *subcuneatus*, *R. fruticosus* agg. L., *R. idaeus* L., and *R. parvifolius* L. [75]. Using SSR, cultivated and wild Romanian *Rubus* accessions (*R. idaeus*, *R. caesius* L., *R. plicatus* Weihe and Nees, *R. plicatus* subsp. *opacus*, *R. hirtus* W. and K., *R. discolor* Focke, *R. sulcatus* Vest ex Tratt., *R. saxatilis* L., *R. suberectus* G. Anders. ex Smith, *R. phoenicolasius* Maxim., *R. occidentalis* L., and *R. spectabilis* Pursh) were compared with some widely grown, advanced cultivars established in Europe and the USA [76]. Significant genetic differences were found between related species and advanced cultivars, indicating a potential value of wild plant material for germplasm diversification.

*Sorbus* and *Crataegus*: Some species in the genera *Sorbus* and *Crataegus* are cultivated as minor crops and are grown regionally in relatively small volumes for special purposes and products. Research has been limited and the process of domestication is still ongoing. Studies of wild populations are important for the selection and introduction of the most suitable plant material. Thus, 302 and 441 genotypes from 17 wild European populations of *Sorbus domestica* L. were analyzed using nuclear and chloroplast SSR loci, respectively [77]. The high genetic diversity may derive from transfer of material between locations. High diversity was observed also in a study of *Crataegus*; 55 accessions of 7 *Crataegus* species grown in China together with European and American reference samples were analyzed with 20 SSR loci selected from hawthorn transcriptional data [78]. Two groups were identified in the Chinese material: one related to European species, and another related to American species. A larger number of *Crataegus* accessions were analyzed in Turkey using selected apple and pear SSR loci [79], while 201 Iranian local genotypes were analyzed by ISSR and SCoT markers [80].

#### 3.1.2. Citrus

Most *Citrus* species have probably originated in southeastern Asia but are now cultivated around the world in warm-temperate to tropical climates, e.g., citron (*C. medica* L.), clementine (*C*. × *clementina* hort. ex Tanaka), grapefruit (*C*. × *paradisi* Macfad.), lemon (*C*. × *limon* (L.) Burm. f.), lime (*C*. × *aurantifolia* (Christ.) Swingle), mandarin (*C*. *reticulata* Blanco), pummelo (*C*. *maxima* (Burm.) Merrill), sour orange (*C*. × *aurantium* L.), and sweet orange (*C*. × *sinensis* (L.) Osbeck). Three of these appear to be ancestral (*C*. *maxima*, *C*. *medica,* and *C. reticulata*) while the remainder have a hybridogenic origin as demonstrated by genome sequencing [81]. Seedlings are produced sexually (regular fertilization) and/or asexually from the nucellus (adventive embryony). In the hybridogenic crops grapefruit, lemon, and sweet orange, true-to-type plants are therefore obtained both by seed and by budding or grafting, and there is generally very little diversity among cultivars, most of which have arisen as sports. By contrast, seed set requires fertilization in citrons, pummelos, and traditional (non-hybridogenic) mandarins, which are therefore more diverse. The genera *Papeda* (papedas), *Poncirus* (trifoliate citrus), and *Fortunella* (kumquats) are also included among citrus fruits.

Genetic relationships and diversity in different citrus crops were investigated with 24 SSR loci in 370 mostly sexually derived accessions preserved at University of California, Riverside, USA [82]. Genetic structure analysis indicated five main populations corresponding to mandarins, pummelos, citrons, trifoliates, and kumquat/papedas, with accessions of hybridogenic crops being admixed as expected. The ancestral species *C. maxima* was studied in detail using 31 SSR loci applied to 274 accessions including local landraces from 12 provinces in China [83]. High levels of diversity were revealed as expected since pummelo has been cultivated for at least 4000 years, and new cultivars are easily obtained from selected seedlings. Three major groups of accessions were identified, most likely corresponding to old dispersal routes within China. Although mandarins are referred to the ancestral species *C. reticulata*, modern mandarins have often been developed by interspecific crosses. Various DNA markers including 50 SSR loci, 24 indel markers, and 67 SNPs were analyzed for 191 accessions from germplasm collections of the Instituto Valenciano de Investigaciones Agrarias (IVIA), Valencia, Spain, and the Station de Recherches Agronomiques (CIRAD-INRA), Corsica, France [84]. Genetic structure analysis showed that all the mandarins belonged mainly to *C. reticulata* but numerous accessions showed some introgression from other ancestral species, especially *C. maxima*.

An Illumina GoldenGate^®^ array (CitSGA-1) was developed with 384 SNPs mined from sequence-tagged site (STS) fragments amplified from genomic DNA of eight cultivars representing citrus breeding in Japan [85]. This array was then used to successfully discriminate 98 citrus accessions from Okitsu Citrus Research Station, National Institute of Fruit Tree Science (NIFTS), Shimizu, Shizuoka, Japan. Another array, this time with 1536 SNPs obtained from sweet orange, was used to genotype a collection of 80 citrus accessions from Florida Citrus Arboretum, Winter Haven, USA [86]. The genetic structure analysis identified five major groups corresponding to the three common ancestral *Citrus* species and *Poncirus* and *Fortunella*. A complex genetic background with introgression between both ancestral and hybridogenic species was demonstrated for several important cultivars.

#### 3.1.3. Grapes

Grapes (*Vitis* spp.) constitute the fourth most important fruit crop in the world [87]. Research has been intense, including identification, characterization, and use of genetic resources in breeding. Like for some important field crops, international consortia have been set up to share *Vitis* research information. Both national and international databases with SSR marker information have been developed for the identification of cultivars (e.g., Pl@ntGrape, le catalogue des vignes cultivées en France *Vitis*, http://plantgrape.plantnet-project.org/; Swiss *Vitis* microsatellite database [88]; International Variety Catalog (VIVC), https://www.vivc.de/). An internationally developed set of nine SSR loci is used as a standard for large-scale screening of *Vitis* genetic resource collections [1,89,90]. The SSR methodology developed for cultivated grapes (*V. vinifera* L.) is applied also to other *Vitis* species [91], such as *V. riparia* Michx, *V. rupestris* Scheele, and *V. berlandieri* Planch., which are used as rootstocks and donors of important traits. Combinations of SSR, CAPS (cleaved amplified polymorphic sequences), and SCAR (sequence-characterized amplified region) markers have been used to screen germplasm collections for resistance to downy and powdery mildew, which are the major grape diseases [92].

Genetic research was facilitated substantially by access to the whole genome sequence together with the introduction of SNP technologies. A first version of the *V. vinifera* genome obtained by shotgun sequencing was published already in 2007 [93]. Several large-scale studies have subsequently been carried out on genetic diversity, cultivar identification, and whole genome associations in *Vitis* collections. Following several updates of the genome sequence and creation of the GrapeReSeq consortium, an 18K SNP array was presented for cultivar identification, parental and population structure clarification, genetic diversity assessment, and applicability for GWAS [25]. The 18K SNP array has been used in many instances; analysis of 279 accessions in the French germplasm repository demonstrated the suitability for identification of associations with highly precise phenotyping data [94], while analysis of a core collection of rootstock accessions (including different *Vitis* species and interspecific hybrids) demonstrated great marker transferability [95]. Analysis of 150 wild *V. vinifera* subsp. *sylvestris* and 77 cultivars and non-vinifera accessions, using both the 18K SNP array and nuclear and chloroplast SSR loci, revealed considerable hybridization with introgression of *V. vinifera* subsp. *sativa* into subsp. *sylvestris* [96].

SNPlex (Applied Biosystems) and Veracode (Illumina) SNP genotyping with 261 nuclear SNPs was performed on 231 *V. vinifera* subsp. *vinifera* and 27 *V. vinifera* subsp. *sylvestris* accessions [97]. The results indicated gene flow between subspecies and supported previous findings of secondary domestication events in Portugal. GBS-derived SNPs were used to screen a set of 472 *Vitis* accessions representing 48 different species from a wide geographic distribution [98]. Analyses of genetic diversity suggested the occurrence of a recent expansion and contraction of effective population size in domestic grapevines, and showed that cultivars from the Black Sea region were quite different from cultivars elsewhere.

#### 3.1.4. Other Temperate Fruit and Berry Crops

Olive, *Olea europaea* L., is an ancient crop with clones that have been multiplied by vegetative propagation for centuries. Domestication is, however, still an ongoing process and seed propagation is common in traditional agroecosystems in the Mediterranean basin (MB). A dataset of 289 different cultivars maintained at World Olive Germplasm Bank of Córdoba, Spain, and two sets of together 96 wild accessions (*Olea europaea* subsp. *europaea* var. *sylvestris*) were genotyped with 25 SSR loci [99]. The cultivated olives were shown to cluster in three different gene pools, corresponding loosely to the west, central, and eastern MB, respectively. The basis of present-day olive germplasm seems to include large-scale admixture and local domestication events.

In a study of French olive germplasm, 113 accessions in the Olive Germplasm Bank, Porquerolles, France, were characterized with 20 SSR loci [100]. For comparison, corresponding genotyping was carried out for 416 accessions originating from 13 Mediterranean countries and preserved in the World Olive Germplasm Bank of Marrakech, Morocco. High genetic diversity was observed among local French varieties, indicating a high admixture level, with an almost equal contribution from the three main Mediterranean gene pools. Some French cultivars were shown to be synonymous with Italian and Spanish cultivars, and a high number of parent–offspring relationships were detected.

Persimmon: The temperate-tropical genus *Diospyros* contains about 500 species, the most important in cultivation being the hexaploid *D. kaki* L.f. native to China but grown extensively in Asia and in other Mediterranean climate regions. Genetic diversity and relationships among 129 accessions originating from China and Japan were investigated with 17 SSR loci [101]. High polymorphism was revealed with considerable differentiation between Chinese and Japanese accessions. In another study, 228 persimmon accessions from five different regions in China were investigated with 12 SSR loci [102]. The five groups produced by a cluster analysis contained accessions from different regions, suggesting considerable gene flow. Similarly, a genetic structure analysis indicated little differentiation between regions.

Vaccinium: Several species in the genus *Vaccinium* have been domesticated recently. Cultivated blueberries belong to North-American species in the section *Cyanococcus*, with the tetraploid *V. corymbosum* L. cultivated in temperate regions around the world. A total of 367 blueberry samples (mostly obtained from the National Clonal Germplasm Repository (NCGR), Corvallis, USA, representing *V. corymbosum*, *V. angustifolium* Ait., *V. virgatum* Ait., and interspecific hybrids) were genotyped with one or both of two multiplex reactions with SSR loci, each consisting of five loci with trinucleotide or longer repeats [103]. The results allowed detection of 54 true-to-type (TTT) cultivars, 13 sets of homonyms (same name but different genotypes), and ten groups of synonyms (different names but the same genotype). Parentage analysis confirmed pedigrees for 54 accessions and identified five of the TTT cultivars among the homonyms and one among the synonym sets.

The diploid North American cranberry (*V. macrocarpon* Ait.) is cultivated on large bogs where accidental seedling establishment may go unnoticed. A total of 271 cranberry plants from 77 accessions representing 66 named cultivars in the NCGR, were investigated with 12 SSR loci to identify TTT material and assess cultivar integrity and relatedness [104]. Sixty-four unique genotypes were found, and some cultivars were shown to contain morphologically similar but not genetically identical “subclones”. The diploid lingonberry (*V. vitis-idaea* L.) is still harvested mostly from wild shrubs in Northern Hemisphere forests, and there has been little genetic research. Still, 1586 GBS-derived SNPs were developed to genotype 56 Canadian samples and find correlations with six environmental variables and two phenotypic traits: total phenolic content and antioxidant capacity of fruit [105].

#### 3.1.5. Subtropical and Tropical Fruits

Avocado: Although the avocado, *Persea americana* Mill., probably originated in Mexico, it is now widely cultivated in subtropical and Mediterranean climate regions around the world. A collection of 100 avocado accessions in the Israeli germplasm bank were genotyped using a set of 109 SNPs previously selected for their ability to detect high levels of polymorphism [106]. A PCA placed the three races (Guatemalan, Mexican, and West Indian) in distinct clusters while genotypes originating from the same breeding program or having the same skin color showed closer relatedness.

Kiwifruit: For many years, the international kiwifruit trade was completely dominated by the *Actinidia deliciosa* (A.Chev.) C.F.Liang and A.R.Ferguson variety “Hayward”, which originated in New Zealand from Chinese plant material. The germplasm used in breeding has increased considerably and now involves a wider range of genotypes including other *Actinidia* species. A set of almost 35,000 GBS-derived SNPs were used to investigate genetic diversity in 89 kiwifruit accessions maintained in the germplasm repository in Namhae Branch of National Institute of Horticultural and Herbal Science, Korea [107]. A cluster analysis placed these accessions in two groups dependent on the presence or absence of hairs on the fruit. A genetic structure analysis instead divided the accessions into five groups; *A. chinensis* Planch., *A. deliciosa*, *A. eriantha* Benth. with wild accessions, female *A. arguta* (Sieb. and Zucc.) Planch. ex Miquel, and male *A. arguta*.

Mango: Several mango species are grown in South Asia but the most important, i.e., *Mangifera indica* L., is now cultivated in tropical areas around the world. Fourteen SSR loci were used to screen 387 mango accessions collected from different regions in India and preserved at the Field Gene Bank at ICAR-Indian Institute of Horticultural Research in Bangalore, and ICAR-Central Institute of Subtropical Horticulture in Lucknow [108]. Two strongly differentiated groups were identified, with accessions originating either in the south-western part of India or in the north-eastern part. A subset with only six SSR loci having very low PI-values (probability of identity, i.e., probability of two cultivars sharing the same genetic profile by chance) were recommended for further use in mango genotyping.

Banana: Originating in South-Eastern Asia/Western Oceania, bananas are of major importance in the tropics. They are usually propagated vegetatively since edible fruits lack seeds. Both dessert and cooking bananas are diploid, triploid, or more rarely tetraploid, and are composed of either genome A (*Musa acuminata* Colla), or A in combination with genome B (*M. balbisiana* Colla). The most popular types of banana are triploid such as the “Cavendish” clones (AAA) and various plantains (AAB). Breeding programs focus mainly on creating triploid and, to an increasing extent, tetraploid cultivars. Fully or partially fertile diploid genotypes are used to create variable progeny, thus emphasizing the importance of access to well-described wild or semi-wild germplasm.

A total of 695 accessions, most of which were obtained from the world’s largest ex situ *Musa* germplasm collection at the International Transit Centre (ITC) in Leuven, Belgium, were genotyped with 19 SSR loci [109]. Overall, cluster analysis-based groups conformed to the morphological traits-based classification. Several samples in need of additional verification were also identified. A total of 591 accessions, with coherent classification and genotyping data, were appointed as a core set (CS) to act as a source with which subsequently analyzed material can be compared. In another study, Sardos et al. [110] genotyped wild and cultivated accessions collected in the Bougainville region (AROB), Papua New Guinea, using the same 19 SSR loci, and subsequently aligned the profiles to the CS database. Joint analysis of CS and AROB data enabled confirmation of the classification of AROB accessions. Duplicates and near-identical genotypes (most likely derived by somatic mutation) could be identified, and gene flow, foreign introductions, and different diversification processes could be hypothesized. A set of 575 Musa accessions from ITC (mentioned above) were analyzed also with DArT markers [111]. PCoA and genetic structure analyses produced groupings that were largely consistent with taxonomical origins and with previous SSR analyses conducted on the same material. However, in contrast to SSR, DArT markers were also able to reveal the genomic composition.

#### 3.1.6. Nut Crops

Economically important nut crops are vegetatively propagated and represent a group of botanically unrelated species. Almond, which is the most widely grown nut crop [87], is mentioned above in the section on *Prunus*. Other important nut crops are walnut (*Juglans regia* L.), cashew (*Anacardium occidentale* L.), pistachio (*Pistacia vera* L.), and hazelnut (*Corylus avellana* L.) [87].

Walnut: In walnut, large-scale genotyping and genetic research has been promoted significantly by the development of an Axiom^®^ 700K SNP array [31] and the assembly and annotation of a reference genome [112]. A study of 95 phenotypically and genetically characterized wild-type accessions demonstrated high genetic diversity in one of the primary centers of walnut origin, and a GWAS of nut- and kernel-related traits was carried out [113]. Efficiency of the SNP array and the SSR loci was investigated in 150 walnut accessions, yielding similar results for creation of a core collection, determination of synonymy and redundant accessions, and genetic structure of germplasm, but SNPs were superior for conducting a GWAS [114].

Cashew: Despite its importance, intra- and interspecific genetic structure has been sparsely studied in cashew. Genetic diversity was analyzed using ISSR markers in 240 wild-type accessions [115], while SSR loci were used to identify endangered populations and develop conservation strategies in the related species *A. humile* St.-Hil. [116]. SNP markers developed in mapping populations are likely to facilitate studies on cashew germplasm in the future [117].

Pistachio: A relatively small number of genotypes have been analyzed in pistachio germplasm studies, based mostly on SSR [118,119], ISSR, IRAP, and REMAP markers [120]. Establishment of a reference genome draft would be very valuable for analysis of genome evolution, identification of genome similarities, and detection of gene clusters associated with stress adaptation as already achieved in other species [121].

Hazelnut is native to Europe and Western Asia, where it has been cultivated for a long time. A GBS-based analysis of the European hazelnut collection with 227 accessions, produced a detailed analysis of the germplasm including identification of genetically distant populations [122]. Accessions of wild *C. avellana* populations may be very valuable for introduction of agronomically important traits, and for a better understanding of crop development [123].

#### 3.1.7. Tuber Crops

Potato: Tuber crops and especially potato (*Solanum tuberosum* L.) belong to the economically most important food sources in the world. Genetic research on potato has, however, been restricted because of polyploidy and high levels of heterozygosity [124]. Nevertheless, genotyping has become an important tool for the identification of potato cultivars and protection of Plant Breeders’ Rights; data for nine SSR loci are included in The European Union Common Catalogue (EUCC) together with 41 phenotypic traits from the International Board for Plant Genetic Resources (IBPGR) descriptor list [20]. Various molecular markers have been used to analyze taxonomic/phylogenetic relationships and genetic diversity among potato accessions [125] including, e.g., inter-primer binding sites (iPBS) for development of breeding populations for parental selection [126]. Potato germplasm research has been enhanced significantly by high-throughput sequencing technologies and NGS, which have resulted in whole-genome sequencing [127] and further development of powerful SNP genotyping platforms as applied in *S. tuberosum* [128] and other polyploid *Solanum* species [129].

Recent threats to cultivated potato like climate change and pathogen pressure have increased the interest in wild relatives, related species and native landraces with a potential for providing valuable sources for breeding programs [130,131,132]. Diploid *Solanum* species are of interest, and some like *S. chacoense* Bitter are already used in potato breeding [133]. To make good use of this germplasm, species population structure, available diversity, and taxonomic affiliation should be investigated [134]. Sequencing of potato wild relatives has resulted in SNP marker sets that have been used to identify resistance genes with potential applications in breeding [135].

Cassava: The allopolyploid and highly heterozygous species *Manihot esculenta* Crantz produces a valuable staple crop in the tropics. Elite cultivars, many of which are triploid, are often both female- and male-sterile, and plants are propagated only by cuttings. Landraces are, however, usually polyclonal since seed propagation occurs, sometimes intentional and sometimes by accident in farmers’ fields. A set of 1580 genotypes belonging to the international cassava germplasm conserved at Embrapa Mandioca and Fruticultura in Cruz das Almas, Bahia, Brazil, were analyzed with 20,601 GBS-derived SNPs [136]. A DAPC revealed 22 groups but these were not related to clustering based on phenotypic data (root color and dry matter) or the genetic and geographic origin.

Sweet potato: The sweet potato (*Ipomoea batatas* (L.) Lam.) is a hexaploid and self-incompatible crop. Traditionally, sweet potato is propagated by cuttings but virus-free tissue culture-derived material has become increasingly desirable due to the high virus pressure [137]. Another approach is to look at diverse germplasm as a possible source of resistance to viruses [138]. SSR markers have been used for the genotyping of about 200 accessions [139], while both SSR markers and phenotypic characterization have been applied in smaller studies [140]. SNP markers have been developed using the specific locus amplified fragment sequencing (SLAF-seq) method to create a core collection from 197 accessions of sweet potato in China [141].

Yams: The genus *Dioscorea* contains approximately 644 species, 12 of which are grown for human consumption in tropical areas around the world, e.g., *D. alata* L., *D. cayenensis* Lam., and *D. rotundata* Poir. In a study of 384 *D. alata* accessions screened with 24 SSR loci, high genetic diversity was apparently associated with chromosome levels and with geographic origin (South Pacific, Asia, Africa, and the Caribbean) and morphoagronomic characteristics [142]. A set of 643 *D. alata* accessions from four continents were subsequently analyzed with 6017 GBS-derived SNPs [143]. Diploid genotypes were more frequent than triploids and tetraploids according to ploidy determinations from allelic frequency distribution at heterozygous loci. Diploids originating from the same geographic area usually clustered together, and polyploidization appears to have occurred separately within each diploid gene pool. A strong domestication bottleneck was suggested, followed by thousands of years of vegetative propagation and polyploidization. This domestication probably occurred independently in the two main gene pools in Asia and the Pacific.

#### 3.1.8. Onions

Onions (*Allium cepa* L.) and garlic (*A. sativum* L.) are widely grown, and breeding and development of new cultivars is carried out as well as conservation and research on genetic resources. Approximately 25,000 accessions of *Allium* are stored in various germplasm collections around the world [144]. The main limitations to the traditional breeding in *Allium* crops are the biannual life cycle, high levels of cross-pollination, and inbreeding depression in onion, and the vegetative propagation and lack of flowering in garlic [145]. In addition, *Allium* has the largest genome among vegetable crops, which complicates sequencing and assembly of genomes, development of markers, genetic mapping, and genomics-assisted breeding [145]. Despite these limitations, whole genome sequencing has been performed for garlic [146], and several molecular markers are applied in germplasm research, like ISSR [147], SSR [148], SNP [149], and DArTseq [150]. Main research interests include assessment of genetic diversity in germplasm collections, identification of accessions, and the development of representative core collections. Population diversity has been analyzed in the material, and the relationship between morphological and genetic differences has been assessed.

#### 3.1.9. Ornamentals

A large number of woody ornamentals are vegetatively propagated and many have large economic importance, although we will only mention two of them here, roses and chrysanthemums. The genus *Rosa* contains approximately 140–180 species, several of which are cultivated as ornamentals while a few, e.g., the dogroses in Section *Caninae* are grown as medicinal plants and as rootstocks. The most commonly grown roses for gardens and the cut-flower trade are interspecific hybrids treated as *R*. × *hybrida* Vill. These roses are highly heterozygous and usually tetraploid with mainly tetrasomic inheritance.

Diversity studies in roses are generally based on SSR loci. Allele copy number has been quantified by scoring the obtained fragment profiles with MAC-PR (multiple allele counting–peak ratio) using high-quality SSR loci [151]. Usually, alleles are instead just scored as the presence/absence, e.g., in a study of 138 rose cultivars analyzed with 24 SSR loci; cut roses, garden roses and rootstock varieties formed significantly different clusters [152]. This study revealed differentiation also between horticultural groups of garden roses (Floribunda, Hybrid tea, Renaissance, etc.), and between cultivars produced in different breeding programs that emphasize different traits and therefore also use different germplasm. A set of 109 Indian-bred Floribunda and Hybrid tea cultivars maintained at the Indian Agricultural Research Institute, New Delhi, India, was genotyped with 31 SSR loci and phenotyped for 59 morphological traits obtained from the DUS guidelines [153]. Cluster analyses produced unclear groupings with both data sets, and a genetic structure analysis with eight groups showed very high levels of genomic admixture as expected. The largest rose diversity study published so far was carried out with 32 SSR loci on 1228 garden roses from 10 rose gardens in France, representing a wide range in the history of rose breeding with special emphasis on France [154]. A DAPC revealed 16 genetic groups, partially consistent with the horticultural groups. Old European and old Asian accessions differed considerably, whereas modern cultivars were more similar, presumably due to an increased utilization of Chinese germplasm in Europe during the 19th century.

Koning-Boucoiran et al. [30] made use of RNA-Seq libraries from tetraploid cut and garden roses, and from diploid *R. multiflora* Thunb. to identify SNPs within and between rose varieties. SNPs, which detected variation among tetraploid roses, were selected for constructing the WagRh SNP 68K Axiom^®^ genotyping array, which has since been used to genotype 96 ornamental rose cultivars for use in several GWAS [155,156,157,158].

Several perennial species of the genus *Chrysanthemum* are important as ornamentals or medicinal plants and are propagated by cuttings; chrysanthemum ranks second after rose in the cut flower trade. Many chrysanthemum species are highly polyploid and have large genomes, which make them less amenable for genetic analyses. A worldwide collection of 159 cut-flower cultivars of *C. morifolium* (Ramat.) Hemsl., maintained at Nanjing Agricultural University’s Chrysanthemum Germplasm Resource Preserving Centre in Nanjing, China, were genotyped using 707 informative fragments derived by SRAP, SCoT, and EST (expressed sequence-tagged)-SSR amplicons [159]. A PCoA differentiated most of the cultivars according to inflorescence type, and to some extent also provenance. By contrast, a genetic structure analysis indicated very low differentiation between two groups, with no association to other traits or origin.

### 3.2. Core Collections

Efforts to screen large germplasm collections, particularly for phenotypic traits, are very costly. A restricted number of accessions can be appointed as a core collection for intensified preservation efforts and research. Definition of a core collection usually involves either (1) some kind of stratification with a cluster analysis or (2) various methods to determine genetic uniqueness. Using the M-strategy method (MSTRAT software), Liang et al. [10] deviced a core collection of 55 Italian apple accessions that retained all the 238 SSR alleles detected in 192 unique genotypes. Based on more than 2000 French apple accessions, Lassois et al. [11] developed a large dessert apple core collection of 278 accessions with 90% of the total dessert apple allelic diversity, and a small dessert apple collection of 48 accessions with 71% of the diversity. For cider apples, a core collection with 48 accessions contained 83% of the total cider apple diversity. In pear, Liu et al. [48] used PowerCore software to design a core collection of 88 accessions out of 385, covering all the rare SSR alleles and 95% of all alleles.

Choice of accessions for a core collection is sometimes based exclusively on genetic diversity estimates as in the examples provided above. However, expert knowledge on factors like popularity, prestige, role in breeding history, or presence of phenotypic traits of interest, can also be considered. Urrestarazu et al. [160] evaluated different strategies for selecting a core collection in Swiss pears, among a total of 841 different genotypes as defined by 15 SSR loci. A subset containing 86 “priority” genotypes chosen by the collection stakeholders was shown to contain two thirds of the SSR alleles present in the entire material, whereas some of the molecular-based strategies were able to retain up to 90% in similar-sized subsets. Overall, strategies involving mixed approaches (stakeholder knowledge and SSR data) did, however, not differ much from purely analytical procedures.

### 3.3. DNA-Based Detection of Synonyms and Clones/Sports

In crops that have been propagated for a very long time, like apple, pear, grapes and olive, the same cultivar names often turn up in different plant collections. Genotyping often shows that these accessions are not genetically identical. In some cases, the discrepancy is probably due to a chance seedling resembling one of its parents but in other cases there are more wide-ranging problems with mistaken identity and mislabelings. Similarly, it is common to find the same genotype under different names in different collections, sometimes due to mistaken identity or the wish for a more “native-sounding” name for a foreign cultivar.

The tried-and-true Jaccard’s similarity coefficient, S_xy_ (= 2n_xy_/(n_x_ + n_y_)), is often used to determine identity by descent (IBD) and thus distinguish between identical and non-identical genotypes. The critical value of course varies between studies depending on methods and material; an IBD value of 0.90 was used as a cut-off point between non-clonality and clonality (i.e., resulting from the same recombination event and thus genetically identical except for possible minor somatic mutations) for apple cultivars analyzed with DArT [12] or GBS [13], while a value of 0.85 was used in another GBS study [37]. IBD values of 0.4158–0.5625 were found for certified first-degree relations in apple [13]. In olive, genotypes are classified as being clonal if they differ by only one or two SSR alleles, and a “reference genotype” is chosen, based on stone morphology and the comparison of SSR profiles of several trees originating from different nurseries and orchards [100].

In polyploid crops, SSR profiles are generally scored as “allelic phenotypes” based on the presence of alleles in each locus but not the (perceived) actual number of copies of each allele. The likelihood of actually overlooking alleles becomes higher with increasing levels of ploidy, which affects the threshold for perceived genotype identity. In a study of hexaploid plum accessions, the cut-off IBD value for clonality was set at 0.88 [60].

### 3.4. DNA-Based Investigation of Pedigrees 

Correct pedigrees are valuable for applied breeding programs but many cultivars in, e.g., root and tuber crops are still derived mainly from open pollination and polycross blocks [161]. Pedigree information is also a backbone for research on trait heritability, estimation of breeding values and conducting marker association analyses. Several studies have therefore aimed to identify relationships among accessions in plant genetic resource collections, build multigenerational pedigrees when possible, and to identify influential cultivars. These investigations usually start with finding the single most likely parent in the collection followed by examination of genotype incompatibilities between offspring and two putative parents based on Mendelian inheritance rules and evaluation of differences in LOD (log-likelihood ratio) between related and unrelated relationships, as described by Khadari et al. [100] in an analysis of pedigrees in olive germplasm.

Using SSR, pedigrees were determined for numerous French apple cultivars, with frequent occurrence of “King of the Pippins” and “Calville Rouge d’Hiver” as founders [11]. Pedigree analyses conducted for Danish apple cultivars at the Pometum, Copenhagen, showed that the English “Cox’s Orange Pippin” and the French “Pigeonnet blanc d’hiver” dominated as parents [9]. Using a SNP array, a corresponding study was conducted in a Swedish heirloom cultivar collection at Alnarp, producing 18 trios (complete parent-offspring families) and several additional parent–offspring relationships often involving foreign cultivars as parents [46].

Screening the apple collection at Brogdale, Great Britain, with DArT markers revealed several parent–offspring relationships including some important triploid cultivars originating in the 18th century [12]. Some of these triploids had diploid parents that possibly date as far back as the 13th century. For a set of 1005 GBS-screened *M.* × *domestica* accessions at USDA, Geneva, USA, “Golden Delicious” had the highest number of first-degree relatives (66) followed by “Red Delicious” (61) [13]. More than half of the accessions were shown to belong to a single extended pedigree. Moreover, 129 accessions had a single-degree relationship with at least one of the current top eight cultivars in USA, suggesting that apple diversity is becoming increasingly restricted and therefore also more vulnerable to environmental change.

In the largest apple pedigree analysis presented so far, information was generated with the Axiom^®^ Apple480K Array for 1425 mostly old European apple cultivars [47]. IBD was calculated among all possible pairs of samples, and exclusion tests were designed based on the number of Mendelian-inconsistent errors. More than 1000 parent–offspring relations and 295 trios were identified. A grandparent couple could be identified for the missing parental side of 26 parent–offspring pairs. Among the 407 parent–offspring relations without a second identified parent, 327 could be oriented because one of the individuals was an offspring in a complete family or by using historical data on parentage or date of recording. The overall pedigree ranged over seven generations and revealed a major impact of two Renaissance cultivars, namely “Reinette Franche” and “Margil”, and one cultivar from the 1700s, “Alexander”. Some of the reported pedigrees had been identified previously with SSR in the same samples [44] but SNP analysis was more informative. Moreover, the problem with more or less sterile triploid cultivars erroneously pointed out as seed parents of diploid cultivars, could be solved only with SNPs. As an example, the triploid “Ribston” is not the mother of “Cox’s Orange Pippin” (as previously suggested from SSR data by, e.g., [9] but instead both cultivars are derived by seed from “Margil”, which has contributed an unreduced eggcell to “Ribston”.

Data obtained with the Axiom^®^ Pear 70K array were used to reconstruct a pedigree network for a total of 1890 diploid samples in the USDA-ARS *Pyrus* collection [50]. Five F1 populations were used to identify erroneous SNPs based on Mendelian inheritance and then aid in the pedigree reconstruction of the *Pyrus* accessions. SNPs were filtered for missing values and then a Mendelian test was run on the known trios in the F1 populations and in the genebank samples, using trio.check as described in Montanari et al. [32], and SNPs with an error rate > 5% were removed. The pruned marker dataset was used to evaluate the relationship between each pair of samples with a robust method (software KING; [162]). In total, 139 previously known trios/duos were confirmed and 498 new ones were identified. A restricted number of ancient and commercially important cultivars like “White Doyenne” (= “Doyenné Blanc” and “Pera Ghiacciuola”) and its offspring “Bartlett” (= “Williams”) are very influential in the *P. communis* germplasm.

SSR genotyping data and available pedigree information was used to check trueness-to-type of 39 cultivars in the German *Rubus* collection [73]. SSR fingerprinting data confirmed female parents of 22 cultivars, male parents of 12 cultivars, and both parents were confirmed for 9 cultivars.

Pedigree relationships were detected with the triadic likelihood estimator (TrioML) in olive accessions collected from all over the Mediterranean basin, suggesting that most first-degree relationships involve cultivars from the same major geographic group [99]. In keeping with these findings, but in contrast to the large influence of foreign cultivars in some of the apple pedigrees mentioned above, SSR analyses in French olives showed that both parents were usually present in the local germplasm, indicating a very active farmer-driven selection program [100].

## 4. Phenotypic Traits

Phenotyping of PGR is generally undertaken for one (or several) of three major reasons; (1) discrimination among accessions and ascertainment of TTT status, (2) identification of potentially valuable traits for use in plant breeding, and (3) genome-wide association studies (GWAS) for development of genetic markers for marker-assisted breeding (MAB) and marker-assisted selection (MAS).

National and publicly funded PGR collections are often screened for the IPGRI (International Plant Genetic Resources Institute) or UPOV (International Union for Protection of New Varieties) descriptors, which were designed to discriminate among different genotypes. Some of these traits are qualitative and possibly controlled by one or a few major genes whereas others are quantitative and have a more complex genetic background. Plant breeding progress is, however, seldom focused on descriptor traits but instead on traits like yield, precocity, flowering and ripening time, storability, disease resistance, and content of various chemical compounds that affect color, texture, aroma and consumer health. Most of these traits are quantitative, i.e., show a continuous distribution, and are controlled by a combination of several quantitative trait loci (QTL).

Some disease resistances are monogenic and easily identified with molecular markers, e.g., the locus-specific apple scab resistance genes. Unfortunately, these resistances are often race-specific and can be overcome by the pathogen or pest at some stage. Genes for broad and more durable resistance are instead quantitatively inherited. Attention has therefore shifted in resistance breeding, from markers for major genes to the development of markers for QTL.

### 4.1. Phenotyping: Protocols and Experimental Set-ups

Phenotyping of tree and bush crops is usually undertaken by examination of a few trees or bushes (sometimes just one) of each genotype during 1–3 years. Phenotyping of shortlived perennials or tuber crops is instead accomplished by annually planting and phenotyping 10–100 plants of each genotype. Ideally, all phenotyping experiments should be replicated not only during several years but also in several locations, but this is seldom possible due to constraints in time and cost. Multiyear data can sometimes be mined from publicly available phenotypic databases like USDA-GRIN, and used for trait inheritance analysis [163]. Historical data collected by, e.g., breeders and gene bank curators, can also become useful to boost the power of statistical analyses [164,165].

The need for plants at a similar developmental stage when assessing morphological traits is usually self-evident although sometimes difficult to achieve when the only material available consists of already existing field plantations. Plant age, and, e.g., type of rootstock and other environmental factors, may however also affect traits like flowering time, chemical fruit contents, and disease and pest susceptibility. Large-scale phenotyping is sometimes attempted by pooling data collected from different genotypes growing in plant collections at different locations. Such a set-up is always problematic because of the considerable environmental variation due to differences in, e.g., field management, soil type and climate. Very detailed methods for standardized phenotyping of apple were developed in RosBREED I [166]. Correlations between sensory and instrumental fruit texture parameters were high in some instances, and moderate year-to-year repeatability of trait values was observed. Protocols were also developed for strawberry [167], peach [168], sweet cherry [169], and sour cherry [170]. Detailed phenotyping protocols can be downloaded from https://www.rosbreed.org/, and publicly available RosBREED-generated phenotypic data can be accessed in the GDR using the “Search Trait Evaluation” page to query qualitative or quantitative traits.

Phenotyping for disease resistance often involves experimental inoculations with, e.g., fungal spores or mycelia, or with spores of bacteria or virus. Large sets of inoculation-based assessments of susceptibility against various fungal storage rots, e.g., blue mold (*Penicillium expansum* Link), bitter rot (*Colletotrichum* spp.), and bull’s eye rot (*Neofabraea* spp.) have been carried out in collections of cultivated apples and wild germplasm (review in Nybom et al. [171]). Correspondence between studies where the same genotypes have been inoculated with the same fungus is often rather low, suggesting that environmental variability during tree growth and fruit development and during the inoculations (including source of inoculum) and evaluations play a large role despite carefully developed phenotyping protocols [172].

As part of RosBREED 2, disease resistance was evaluated and standard protocols developed for inoculation and evaluation of, e.g., fire blight (*Erwinia amylovora*) in apple [173], brown rot (*Monilinia* spp.) in peach [174], and cherry leaf spot (*Blumeriella jaapii* (Rehm.) Arx) in sour cherry [175]. In stone fruits, accurate scoring of resistance against plum pox virus (“sharka”) is very valuable for further research [176].

### 4.2. Associations with Genetic Markers

Until now, genomic regions associated with a trait of interest have usually been identified from information on marker segregation in biparental mapping populations. Development of such populations is, however, expensive and time-consuming when working with long-lived perennial crops. Many QTL mapping studies in, e.g., apple have been performed with relatively small sample sizes, making it difficult to detect small-effect QTLs that cause variation in complex traits [177].

In order to improve the QTL validation procedures, reference germplasm sets were developed within the RosBREED project for apple, peach, sweet cherry, and strawberry [167,178]. These sets contained designated important breeding parents (IBPs) based on average marker allele representation in relatives. A standardized phenotypic evaluation of the reference germplasm sets was conducted during several years across the multiple breeding programs participating in RosBREED [4]. This approach made it possible to combine data sets across programs for improved statistical power to ensure that trait–DNA associations could be accurately determined.

Although MAS functions reliably for monogenic and oligogenic traits controlled by a few QTLs with medium to large effects (individual loci explain at least 10% of the variance), occurrence of polygenic traits (many loci, each with a very small effect) might be more appropriately addressed using genome-wide association studies (GWAS) or genomic selection (GS). 

### 4.3. Genome-Wide Association Studies (GWAS)

GWAS was developed for the detection of QTLs based on associations between genome-wide markers and trait phenotypes. A large number of unrelated genotypes is needed, e.g., germplasm collections, since these can produce a much higher mapping resolution than classical offspring families. In addition, the number of QTLs that can be identified for a given phenotype is not limited by the segregation products of a specific cross, but rather by the number of QTLs underlying the trait and the genetic diversity of the studied germplasm collection.

Apple and pear: Two different approaches were used in a study of fruit texture in apple; (1) a pedigree-based analysis (PBA) using six full-sib pedigreed families, and (2) a GWAS based on a collection of apple accessions, thus making it possible to exploit a larger range of both genetic and phenotypic variation [179]. Both plant materials were genotyped with a 20K SNP Infinium^®^ array and phenotyped with a high-resolution texture analyzer. QTL mapping was undertaken using the seedling families, while a GWAS performed on 233 accessions produced detailed information on associations between allelic configurations and associated traits. A set of 172 apple accessions from the Nova Scotia Fruit Growers’ Association Cultivar Education Trial, Kentville, Canada were genotyped with GBS-derived SNPs and phenotyped during two years for fruit quality traits and resistance/susceptibility to apple scab (*Venturia ineaqualis* (Cooke) G.Winter) [180] and for content of phenolic compounds [181]. GWAS revealed several known loci for skin color, harvest date, and firmness at harvest. Several significant GWAS associations were also detected for resistance to scab but these hits are likely to represent a single ancestral source. In addition, GWAS proved to be useful for identification of candidate genes to produce several chemical compounds. A 200K Axiom^®^ pear SNP array was applied for screening 188 variable pear accessions and an F_1_ population with 98 individuals [33]. A GWAS revealed one gene associated with flowering time and several candidate genes linked to fruit size.

Table (dessert) apples differ considerably from cider apples in traits related to fruit quality and taste. Two sets with 48 dessert and 48 cider apple accessions, respectively, in the INRA Angers germplasm collection were genotyped with the 8K Illumina^®^ SNP array [182]. The genome-wide level of genetic differentiation between cider and dessert apples was low, but 17 candidate regions showed evidence of divergent selection, displaying either outlier F_ST_ values or significant results in a GWAS when bitter versus sweet fruits (as denoted in the literature) were compared. Historical phenotypic data (downloaded from the GRIN-USDA database) were used in a study of 689 apple accessions at USDA, Geneva, New York, screened with 4395 GBS-derived SNPs [163]. Significant genetic differentiation was found between Old World and New World cultivars while a GWAS of 36 phenotypic traits confirmed an already known association between fruit color and the *MYB1* locus, and novel associations between the transcription factor *NAC18.1* and harvest date and fruit firmness. In another study, comparison of genome sequences of *M.* × *domestica* and *M. sieversii* suggested that domestication might have altered the frequency of genes related to stress response [13]. Differentially enriched genes in dessert apples and in cider apples, respectively, were also compared, and a GWAS indicated strong selection for red skin color (*MYB1* gene) and possible selection for firm fruits (*NAC18.1*) during domestication.

Not just texture and sugar/acid ratio but also aroma is important in apple breeding. The entire apple volatilome and fruit texture was phenotyped in 162 apple accessions, and a GWAS was performed using a 9K SNP array [183]. Marker–trait associations were identified and found to colocate with candidate genes for aroma, such as *MdAAT1* and *MdIGS*. Moreover, analysis of allelic configuration of two loci for fruit texture variation, *MdPG1* and *MdACO1*, revealed an interplay between fruit texture and the production of volatile organic compounds. Larsen et al. [165] used GBS-derived SNPs in a GWAS together with data for volatiles screened in the juice of 145 accessions, and for sugar and acid quantity in 110 accessions. Several marker–trait associations were detected, with the most reliable involving butyl acetate and hexyl acetate on chromosome 2 in a region of several alcohol acyl-transferases. Significant SNP associations were found for total sucrose content, and for fructose and sucrose in percentage of total sugars. GWAS using historical data on 64 phenotypic traits recorded in 177 accessions, showed strong associations with fruit color and the correlated traits harvest date, eating time, and earliness.

Within the FruitBreedomics project, approximately 1200 apple genotypes distributed over six locations in Europe (Sweden, Great Britain, Belgium, France, Italy, and Czech Republic) were phenotyped for various pomological characters and then genotyped with the 480K Axiom^®^ SNP array [184]. A set of 10 common cultivars was used to adjust phenotypic values for year and site effects. A GWAS calculated across all six collections retained two SNPs as cofactors on chromosome 9 for the flowering period and 6 for the ripening period (both traits scored on an ordinal scale from 1 to 9), which together accounted for 8.9 and 17.2% of the phenotypic variance, respectively. Variation in genotypic frequency of the SNPs associated with the two traits was connected to the geographic origin of the genotypes (North+East, West, and South Europe), and indicated differential selection in different growing environments.

Stone fruits: Data obtained by screening 1580 accessions of peach and some related *Prunus* species with the peach 9K SNP array were used in GWAS, focusing on traits determined by single Mendelian genes, like fruit pubescence, fruit shape, fruit flesh color, non-melting/melting flesh, titrable acidity, leaf gland type, and showy/non-showy flower type [57]. Some close associations were found but application in MAS is not straightforward and must be validated. In another study, 480 peach genotypes were sequenced and the resulting SNPs used in a series of GWAS (with 165–415 accessions) to reveal associations with flesh color, fruit hairiness, fruit shape, fruit texture, flesh adhesion, pollen fertility, total levels of phenolic compounds, skin color, fruit weight, fruit soluble solids content, and low chilling requirement [55]. This study involved accessions at different levels of domestication, thereby allowing the identification of gene groups that have changed during the breeding process, as well as suggesting future breeding options. GBS-derived SNPs were also used in a GWAS to identify candidate genes for resistance to plum pox virus in 72 breeding accessions of apricot [176].

Grapes: The GrapeReSeq 18K Vitis SNP array was used to screen 783 accessions of various *Vitis* species, and a series of GWAS were applied to study traits like budburst-to-veraison time, cluster weight, muscat flavor, wine acidity, seedlessness, berry skin color, and sex [25]. In another study of only 24 *Vitis* genotypes, a very large number of GBS-derived SNPs were able to reveal associations with berry shape, number of seeds, panicle type, berry sucrose content, berry acid content, and 12 aromatic compounds [98].

Citrus: In a GWAS, 2309 GBS-derived SNPs were used to genotype 110 accessions maintained at Kuchinotsu Citrus Research Station, Nagasaki, Japan, including landraces, modern cultivars and breeding lines evaluated for fruit quality [185]. Seven QTLs could be identified, including four novel ones; four significant QTLs for fruit weight and one each for fruit skin color, pulp firmness, and segment firmness. Historical germplasm data for 111 citrus accessions were combined with data obtained for a set of 676 F_1_ individuals from 35 full-sib families maintained at Institute of Fruit Tree and Tea Science, NARO (NIFTS), Nagasaki and Shizuoka, Japan [164]. The whole material was genotyped for 1841 SNPs on two arrays and evaluated for 17 fruit quality traits. Detection of significant associations was improved using data from all individuals instead of data from only the germplasm accessions.

Banana: A set of 105 accessions were genotyped with 5544 GBS-derived SNPs, revealing high levels of admixture in most accessions, except for a subset of 33 samples from Papua New Guinea [186]. A GWAS showed that 13 genomic regions contained genes that appear to be linked with the seedless phenotype (i.e., parthenocarpy combined with female sterility) including one strong candidate gene for female sterility.

Cassava: The first GWAS in cassava was conducted with 6128 African breeding lines preserved in Nigeria and Uganda. These were phenotyped for susceptibility to cassava mosaic disease in annual field trials for two years and genotyped-by-sequencing for 42,113 reference genome-mapped SNP markers [187]. One region on chromosome 8 was found to account for 30–66% of the genetic variation in susceptibility/resistance to the disease. Another GWAS was conducted with 672 cassava clones preserved at International Institute of Tropical Agriculture (IITA) in Nigeria [188]. These clones were assessed for dry matter content and root yellow color (indicative of carotene content) in annual field trials during three years, and genotyped-by-sequencing for 70,000 SNP loci. Two major loci for root yellowness were identified on chromosome 1, one of which also colocated with dry matter content. A third study was conducted on 158 cassava accessions from the Chinese Cassava Germplasm Resources, originally collected from all around the world and grown with three replications in each of four Chinese provinces [189]. The accessions were phenotyped for various agronomic traits scored in annual field trials during a three-year period. Each accession was also genotyped with the amplified-fragment SNP and methylation (AFSM) method yielding a total of c. 350,000 SNPs and indels. Using GWAS, 36 SNPs showed associations with 11 of the agronomic traits.

Ornamentals: An association panel with 96 mostly tetraploid roses was investigated for a number of commercially important traits. Carefully conducted experiments were performed to provide data for concentration of anthocyanins and carotenoids in the rose petals [155], the capacity of leaf petioles to generate adventitious shoots [157], the capacity to form roots in vitro and in vivo [156], and the capacity to form calluses on leaf explants in vitro [158]. This rose panel was also genotyped with the WagRhSNP 68K Axiom^®^ SNP array, and several GWAS were performed. Five genomic regions proved to be associated with the total anthocyanin content, two large clusters were associated with the carotenoid content, and several candidate genes were identified with known functions in either the anthocyanin or the carotenoid biosynthesis pathways [155]. A total of 88 SNPs were associated with either the shoot regeneration rate or the shoot ratio, with 12 SNPs derived from ESTs matching candidate genes involved in shoot morphogenesis [157]. For the root formation, 49 SNPs were associated with in vitro root length, while 98 SNPs were associated with root number in vivo, 218 SNPs with root length in vivo and 4 SNPs with root biomass in vivo [156]. Some of these SNPs were located in genes with homology to root-related genes. In the study on callus formation, a large cluster of markers was found on chromosome 3 and minor clusters on other chromosomes [158]. A rose *APETALA2/TOE* homologue was identified as a major regulator of petal number on chromosome 3 using genetic information acquired in the process of developing a high-quality genome sequence in diploid rose [190]. A few association studies have been carried out also in other ornamentals. Thus, a GWAS was conducted in *Chrysanthemum morifolium* with 707 markers including SNPs, and 11 quantitative morphological traits, producing 54 significant associations that were consistent over the two years of phenotypic data collection [159].

## 5. Concluding Remarks

Some of our conclusions from the review of large-scale studies of germplasm for vegetatively propagated plants are as follows:

Standardization of SSR loci: SSRs are widely applied in all kinds of crops, and they allow cost-effective studies also for smaller number of samples. In addition, they are usually sufficiently diverse for estimation of overall diversity in PGR collections and analyses of genetic structuring, discrimination between sexually and asexually (vegetatively) derived offspring, and pedigree determinations. However, standardization of the loci used should be improved to facilitate the merging of data sets originating from different laboratories. It should also be mentioned that SSRs do not provide sufficient information for fine-scale analysis of marker-trait associations.

GBS-derived SNPs: Due to the fast decreasing costs for genetic sequencing, GBS-derived SNPs are increasingly often applied, even in crops that lack an annotated reference genome. In addition to estimating genetic diversity, etc., these SNPs can also yield important information on marker–trait associations. It is, however, difficult to see how SNP data produced in different laboratories could be merged to create larger data sets.

SNP arrays: Genome sequencing, assembly, and annotation has contributed to the development of large-scale research methods in some of the more important crops. A number of crop-specific SNP array systems have been developed using these data, and are now being applied for assessment of diversity, etc., and are often used also for analysis of marker–trait associations. These SNP arrays have the added benefit of allowing later additions into a database although the set of new samples should be close to 100 for cost effective laboratory analyses.

Phenotyping systems: Modern high-throughput phenotyping systems are available for many crops, concerning, e.g., plant morphology and physiology and chemical contents and disease resistance. Still, the number of phenotyping studies is low and so is the number of accessions involved in such studies. To fully reflect the response of each genotype, data should ideally be collected under different plant management and climatic conditions during several years, but this is seldom achieved. The problems are especially pronounced for tree crops; time and financial constraints seldom allows propagation and planting of a randomized experiment with replicated genotypes. Instead, already existing PGR plantings are often utilized, both for measurements carried out on the plants themselves and for measurements carried out on, e.g., harvested fruit.

GWAS: In some crops, functionally important genes have been identified, and marker-assisted breeding methods have been developed for their introduction into new cultivars. Application of GWAS based on large-scale and high-quality genotyping and phenotyping has the potential to reveal the exact position of key genes on the genome, and further improve molecular breeding methods. Presently, the major obstacle seems to be the lack of sufficiently informative phenotypic data. International multilocus trials of a common set of diverse genotypes could be a very valuable resource in future research.

## Data Availability

Not applicable.

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
