# Peer review of "Recent Large-Scale Genotyping and Phenotyping of Plant Genetic Resources of Vegetatively Propagated Crops"

_plants, 2021, doi:10.3390/plants10020415_

Round 1

Reviewer 1 Report

In this manuscript, the authors gave an overview of the methods for genotyping nine groups and phenotyping of eight groups of vegetatively propagated crops. The authors listed the most important genotyping and phenotyping data of the crops. Such studies were conducted as part of projects that were conducted in many countries on all continents. The studies have produced a large amount of data and therefore standardization of data is essential for their comparison and drawing conclusions that would be commercially useful. The most commonly applied markers are SSRs and SNPs. GWASA enables a new approach to combine genotypic and phenotypic characteristics to improve molecular breeding methods of the cultivars.

The manuscript presents an overview of research concerning genotyping and phenotyping of the most economically important vegetatively propagated crops.

I suggest the following minor changes:

Line 48: Change PGR in Program for Genetic Resources (PRG)

Line 175: Delete: (Zum et al 2020)

Line 335: Italic: P. cerasifera

Line 387: Define abbreviation: ISSR and SCoT (SCoT abbreviation is first used in Line 387, not in Line 727)

Line 445: Define abbreviation: CAPS (Cleaved Amplified Polymorphic Sequences) and SCAR (Sequence Characterized Amplified Region)

Line 575: Delete: (mentioned above)

Line 601: Define abbreviation: Inter-retrotransposon Amplified Polymorphism (IRAP) and Retrotransposon-Microsatellite Amplified Polymorphism (REMAP)

Line 665: Delete: respectively

Line 709: Delete: 30

Line 906: Check the ref. Winslow et al.]

Line 1025: Define abbreviation: Institute of Fruit Tree and Tea Science, NARO (NIFTS)

Author Response

Dear Editor of Plants,

Thanks to the reviewers for taking the time to review our manuscript. We thank you for the good evaluation of the manuscript, as well as the suggestions made for improving its quality.

We essentially followed all suggestions provided by reviewers except the one about providing a separate list with abbreviations. However, instead a separate abbreviation list, we have taken great care to spell out abbreviations throughout the whole manuscript.

Sincerely,

Hilde Nybom

Gunars Lacis

Reviewer 2 Report

Manuscript - review - Recent large-scale genotyping and phenotyping of plant genetic resources of vegetatively propagated crops by Hilde Nybom and Gunars Lacis is a widely contained study. It can be useful for people dealing especially with genetic resources conservation and description.

I have only several comments:

  1. There are a lot of abbreviations without explanation. I recommend adding a list of abbreviations and their meanings before the introduction.
  2. In chapter 3 DArT Seq is mentioned and this method is missing in the chapter 2
  3. In the chapter 3.1, there is only SSR data treatment. Methods how to evaluate data from analyses based on NGS and SNP arrays should be added too - at least shortly.

Author Response

(The authors gave the same response as above.)
